# No Evidence of an Association between Genetic Factors Affecting Response to Vitamin A Supplementation and Myopia: A Mendelian Randomization Study and Meta-Analysis

**DOI:** 10.3390/nu16121933

**Published:** 2024-06-18

**Authors:** Xiaotong Xu, Nianen Liu, Weihong Yu

**Affiliations:** 1Peking Union Medical College, Chinese Academy of Medical Sciences, Beijing 100730, China; pumc_2022xuxiaotong@student.pumc.edu.cn; 2Department of Ophthalmology, Peking Union Medical College Hospital, Chinese Academy of Medical Sciences, Beijing 100730, China; 3Key Laboratory of Ocular Fundus Diseases, Chinese Academy of Medical Sciences, Beijing 100730, China; 4Research Unit of Myopia Basic Research and Clinical Prevention and Control, Chinese Academy of Medical Sciences, Wenzhou 325027, China; 5Fifth School of Clinical Medicine, Peking University, Beijing 100730, China; nianenliu@163.com; 6Department of Ophthalmology, Beijing Hospital, National Center of Gerontology, Institute of Geriatric Medicine, Chinese Academy of Medical Sciences, Beijing 100730, China

**Keywords:** myopia, vitamin A, mendelian randomization, meta-analysis

## Abstract

The relationship between vitamin A supplementation and myopia has been a topic of debate, with conflicting and inconclusive findings. We aimed to determine whether there is a causal relationship between vitamin A supplementation and the risk of myopia using Mendelian randomization (MR) and meta-analytical methods. Genetic variants from the UK Biobank and FinnGen studies associated with the response to vitamin A supplementation were employed as instrumental variables to evaluate the causal relationship between vitamin A supplementation and myopia. Fixed-effects meta-analysis was then used to combine MR estimates from multiple sources for each outcome. The meta-analysis of MR results found no convincing evidence to support a direct causal relationship between vitamin A supplementation and myopia risk (odds ratio (OR) = 0.99, 95% confidence interval (CI) = 0.82–1.20, *I*^2^ = 0%, *p* = 0.40). The analysis of three out of the four sets of MR analyses indicated no direction of causal effect, whereas the other set of results suggested that higher vitamin A supplementation was associated with a lower risk of myopia (OR = 0.002, 95% CI 1.17 × 10^−6^–3.099, *p* = 0.096). This comprehensive MR study and meta-analysis did not find valid evidence of a direct association between vitamin A supplementation and myopia. Vitamin A supplementation may not have an independent effect on myopia, but intraocular processes associated with vitamin A may indirectly contribute to its development.

## 1. Introduction

Myopia is a common refractive error characterized by the focusing of parallel rays of light in front of the retina when the eye is in a relaxed state. Over the past few decades, the global prevalence of myopia has increased significantly, particularly in East and Southeast Asian countries, where rates exceed the global average. Projections indicate that by 2050, half of the global population will be myopic [1,2]. Myopia is a complex condition influenced by both genetic and environmental factors and is also associated with multiple ocular diseases such as retinoschisis, glaucoma, and macular degeneration, potentially leading to permanent vision loss in severe cases [3,4,5]. Therefore, considering the continuous increase in myopia prevalence and its adverse consequences, identifying risk factors and understanding its pathogenesis are crucial for effective prevention and delay of its onset.

An increasing number of scientists have recognized that myopia can be prevented and controlled through dietary management or additional supplementation of certain nutrients. For example, reducing sugar intake can prevent myopia induced by scleral glycolysis, which is activated by elevated postprandial insulin [6]. Additionally, oral supplementation with omega-3 polyunsaturated fatty acids partially alleviated near-work-induced reduction of choroidal blood perfusion, thereby reducing myopia [7]. Recent epidemiologic studies have also shown that vitamins, particularly vitamin D and vitamin A, which are both fat-soluble vitamins, may significantly impact refractive errors such as myopia [8,9,10,11]. Among these, there are more studies on the association between vitamin D and myopia, with the widely accepted hypothesis being that outdoor activity may inhibit myopia progression by modulating vitamin D levels [12]. The findings related to vitamin A are less clear. Some studies found that retinoic acid (RA), the major functional form of vitamin A in vivo, increases in the retina of guinea pigs in models of form deprivation myopia (FDM) and lens-induced myopia (LIM) [13,14] and decreases after positive defocus [15]. In humans, multiple genome-wide association studies (GWAS) have identified SNPs in genes associated with RA signaling, such as retinol dehydrogenase 5 (RDH5), retinal G-protein-coupled receptor (RGR), and RA receptor-associated orphan receptor beta (RORB) [16], suggesting that vitamin A can promote myopia in vivo via RA. However, data from the Raine Study Gen2 indicated that the association between adequate dietary intake of vitamin A during adolescence and the risk of myopia in adulthood was not significant after adjusting for potential confounding factors [10], raising doubts about the efficacy of vitamin A in myopia prevention and control. And there are few studies on the effects of vitamin A supplementation on myopia, whether through dietary intake or additional supplementation. Limitations of the current evidence include the difficulty of simultaneously assessing the effects of multiple dietary components and their long-term effects on myopia. Additionally, relevant studies only estimated dietary intake of vitamin A without considering the potential impact of additional supplementation, and these studies are vulnerable to confounding, reverse causation, and measurement error. Despite growing scientific interest in the relationship between vitamin A and myopia, more robust evidence is needed before recommending supplementation for the prevention or control of myopia.

To this end, we conducted Mendelian randomization (MR) analysis. MR is an innovative and efficient method for causal inference [17]. It utilizes the random distribution and combination of genetic variants during gamete formation to create new randomized populations and employs genotypes as instrumental variables to assess causal relationships. MR utilizes instrumental variables (IVs) constructed from genetic variants with established effects on risk factors to make causal inferences [17]. Theoretically, MR can minimize the impact of confounders, avoid reversed causation, and provide adequate statistical power [18]. In recent years, MR has been widely used to investigate causal relationships of risk factors in recurring diseases, enabling researchers to delve more deeply and precisely into these causal relationships [19,20,21]. Additionally, the use of MR to assess factors influencing eye diseases has gained significant attention. For instance, some MR studies have established causal relationships between vitamin D, blood sugar, sleep, and myopia [22,23,24].

As mentioned earlier, the effect of vitamin A on myopia is unclear. While evidence indicates that vitamin A deficiency can have adverse effects on the eyes [25], there is no direct research supporting a link between additional vitamin A supplementation and myopia. Thus, in our study, we employed MR analysis using genetic variants associated with the response to vitamin A supplementation as IVs to evaluate the relationship between vitamin A supplementation and myopia. Additionally, we conducted a meta-analysis of MR results across different groups to assess the association between vitamin A supplementation and the risk of myopia, aiming to determine causal effects.

## 2. Materials and Methods

### 2.1. Study Design

This two-sample MR study aimed to explore the causal impact of vitamin A supplementation on the risk of myopia. The study utilized data from the publicly accessible UK Biobank study and the FinnGen study. Each cohort was strictly approved by their respective ethics committees, so no additional approval was required for this study. We estimated the association between genetic predisposition to the response to vitamin A supplementation and myopia in FinnGen and UK Biobank cohorts and then performed a meta-analysis to combine the association estimates. Genetic predisposition to the response to vitamin A supplementation refers to the potential impact of these genetic variations on the body’s response to vitamin A supplementation, which might affect its efficacy in preventing or mitigating myopia.

### 2.2. Data Sources

This study included data from the UK biobank and FinnGen, comprising 4 groups for vitamin A supplementation and 4 groups for myopia, respectively. We analyzed vitamin A supplementation as the exposure using 4 datasets, all collected by questionnaire from European cohorts regarding regular or previous day vitamin A supplementation via tablets or capsules. The datasets included ukb-a-458 (*n* = 335,591) integrated by the Neale Lab, available at: http://www.nealelab.is/uk-biobank (accessed on 27 February 2024), ukb-b-9596 (*n* = 460,351) integrated by the MRC IEU [26], “Phenotype: Treatment/medication code: vitamin a” (*n* = 361,141) [27] and “Phenotype: Vitamin and/or mineral supplementation use: Vitamin A” (*n* = 51,427) [26] from the UK Biobank. Myopia data, as the outcome, were also included in 4 datasets: ukb-a-419 (*n* = 335,700) integrated by the Neale Lab, ukb-b-6353 (*n* = 460,536) integrated by the MRC IEU, finn-b-H7_MYOPIA (*n* = 212,571) obtained from the FinnGen consortium and “Phenotype: Which eye(s) affected by myopia (short sight): Both eyes” (*n* = 29,317) from the UK Biobank. Comprehensive details on Single Nucleotide Polymorphisms (SNPs) utilized in this study are provided in Table 1.

### 2.3. Instrumental Variable (IV)

We applied the following criteria to select the IVs: (1) potential IVs were selected based on SNPs associated with each gene that surpassed the locus-wide significance threshold (*p* < 5.0 × 10^–6^); (2) linkage disequilibrium (LD) between the SNPs was calculated using the reference panel from the 1000 Genomes Project European samples data. SNPs with R^2^ < 0.001 (clumping window size = 10,000 kb) were filtered, retaining only the SNPs with the lowest *p*-values; (3) SNPs with minor allele frequency (MAF) ≤ 0.01 were excluded; and (4) in cases where palindromic SNPs were detected, the alleles on the forward strand were determined using information on allele frequencies.

### 2.4. Statistical Analysis

The primary MR method employed was the random-effects multiplicative inverse variance weighted (IVW) method, as described by Burgess et al. [28], to evaluate the associations between genetic predisposition to the response to vitamin A supplementation and the risk of myopia. MR estimates for each outcome from various sources were subsequently combined using the fixed-effects meta-analysis.

We utilized the MR-Egger regression method to identify potential directional pleiotropy by evaluating the significance of the intercept deviation from zero. The MR-PRESSO method was employed to identify and rectify potential outliers, and the MR-PRESSO global test was used to evaluate horizontal pleiotropy arising from heterogeneity among SNPs’ estimates.

We then performed sensitivity analyses using the MR-Egger and IVW methods. And Cochran’s Q statistics were utilized to assess the heterogeneity of IVs [29]. We conducted a “leave-one-out” analysis to identify potentially heterogeneous SNPs and evaluate the influence of each instrumental SNP on the overall results. Additionally, we calculated the F statistic for each SNP to determine the strength of the IVs. An F value exceeding the recommended threshold of 10 indicated no significant weak instrumental bias [30]. All analyses were conducted in R 4.3.2, utilizing packages such as “TwoSampleMR”, “meta”, and “MRPRESSO”.

## 3. Results

### 3.1. Verification of Causality

Genetic liability for the response to vitamin A supplementation showed no association with increased risk of myopia in the meta-analysis of all outcome sources (OR = 0.99, 95% CI = 0.82–1.20, *I*^2^ = 0%, *p* = 0.40, Figure 1A), indicating that vitamin A supplementation is neither a risk nor protective factor for the development of myopia.

Among the four sets of MR analyses, the results MR-Egger, Weighted median, Inverse variance weighted, weighted model, and simple model analyses in three of the sets consistently indicated no significant causal effect (*p* > 0.05, Figure 2). The set of MR results with the highest weighted effect value (90.7% weighted) showed that vitamin A supplementation did not play a role in the development of myopia (OR = 1.002, 95%CI 0.820–1.224, *p* = 0.987, Table 2, Figure 1B). The ORs of the other two sets of results were also very close to 1 (OR = 1.120, 95%CI 0.363–3.455, *p* = 0.844; OR = 0.864, 95%CI 0.405–1.844, *p* = 0.706, Table 2, Figure 1C,D). Another set of results suggested a significant trend in the IVW method, indicating that for every 1-unit increase in vitamin A supplementation (e.g., 1 mg or 3000 IU), the risk of myopia was increased by 0.0019-fold, approaching formal significance (OR = 0.002, 95% CI 1.17 × 10^−6^–3.099, *p* = 0.096, Table 2, Figure 1E). However, the weight of this effect value was the smallest among the four results, accounting for only 0.1%.

### 3.2. Heterogeneity Test

The statistical Q values obtained from MR-Egger and Cochran’s Q test of the IVW method indicated the absence of significant heterogeneity among SNPs (*p* > 0.05, Table 3). Furthermore, the lack of significant difference between the MR-Egger intercept term and 0 (*p* > 0.05) indicated the absence of horizontal pleiotropy among the SNPs (Table 4). The symmetrical distributions observed in the funnel plot provided further evidence that potential bias had a minimal impact on the causal associations observed in the study (Figure 3).

### 3.3. Sensitivity Analysis

Following the leave-one-out test, the analysis results using the remaining SNPs were similar to those obtained when including all SNPs, even after systematically removing each SNP related to vitamin A supplementation. No SNPs were identified as having a substantial effect on the causal association results, suggesting that the findings of this MR study were consistent and reliable (Figure 4).

## 4. Discussion

In this study, we employed a combination of two-sample MR analysis and meta-analysis to investigate the association and causality between vitamin A supplementation and myopia, utilizing summary statistics from the UK Biobank and FinnGen cohorts. Our results showed no evidence of a causal relationship between vitamin A supplementation and myopia. Among the four sets of MR analyses, the MR-Egger, weighted median, IVW, weighted model, and simple model analyses for three of the outcomes indicated no direction of causal effect. However, one set of results suggested that vitamin A supplementation might be a protective factor for myopia, with the results of the IVW method nearly reaching significance (OR = 0.002, 95% CI 1.17 × 10^−6^–3.099, *p* = 0.096, Figure 1). Nonetheless, the weight of this effect size was minimal, accounting for only 0.1%.

While traditional observational studies are often limited by confounding factors, the combination of meta-analysis and MR analysis using SNPs as instrumental variables helps mitigate this issue. MR analysis, by leveraging genetic variants as proxies for the response to vitamin A supplementation, provides stronger causal inference and is less susceptible to confounding and reverse causality biases. This approach enhances the reliability of our findings. Moreover, MR studies typically do not require ethical review, as they rely on existing genetic data. In our study, specific SNPs, such as rs1667255 in the BCMO1 gene, rs10882272 and rs7501331 in the RBP4 gene, and rs10882280 in STRA6, were selected due to their known roles in vitamin A metabolism [31,32]. BCMO1 encodes an enzyme crucial for converting beta-carotene to retinal, influencing vitamin A levels in the body (Ferrucci et al., 2009). RBP is responsible for the storage and transportation of retinol, ensuring the effective use of retinol in the body to ensure the normal function of the retina. The function of STRA6 is essential for the retina and other tissues to obtain sufficient retinol. Its genetic variants may affect the metabolism of retinol, affecting the function of the retina and the process of myopia. Other related SNPs were also taken into account. The interaction between these genetic factors and vitamin A supplementation was considered to elucidate potential mechanisms underlying the observed associations.

Although our analysis showed no association between vitamin A supplementation and myopia, we speculate that vitamin A is theoretically involved in the development of myopia. Vitamin A itself is not the main biologically active intermediary of its functions. As previously mentioned, all-trans retinoic acid (atRA) and 11-cis retinaldehyde are critical regulators of gene transcription related to vitamin A’s functions [33]. Evidence from GWAS studies suggested that retinoic acid metabolism genes contribute to susceptibility to refractive errors [34]. Serum vitamin A levels are closely associated with tissue atRA concentrations [35]. For instance, after supplementing with 25,000 IU of retinyl palmitate (vitamin A) for 360 days, patients receiving PRK for myopia showed significantly accelerated epithelial regeneration (*p* = 0.029), reduced incidence of opacity (*p* = 0.035), and significantly improved uncorrected visual acuity (*p* = 0.043) compared to the control group, with no reported side effects or clinical evidence of chronic hypervitaminosis A [36]. Despite the lack of direct evidence linking atRA concentrations in the retina and choroid to serum vitamin A, Obrochta et al. suggested that a decrease in dietary vitamin A resulted in reduced atRA concentrations in the testes [35]. Similar to the blood–testis barrier, the blood–eye barrier selectively permits the entry of serum components into the eye. Furthermore, oral RA can effectively increase RA levels in eye tissues within 8 h [37]. Therefore, it is plausible that atRA concentrations in the retina and choroid are closely related to serum vitamin A levels, indicating vitamin A supplementation could influence intraocular processes by increasing retinal and choroidal atRA levels.

Several animal studies have shown that RA can serve as a chemical signal regulating eye growth [38,39]. For example, in rhesus monkeys with experimentally induced myopia, the synthesis rate of RA in the retina and choroid/RPE was significantly higher in comparison to the control group (*p* < 0.01) [40]. Increased atRA levels have been found in mammalian FDM and LIM models as well [14,41,42]. Additionally, feeding guinea pigs or chicks with atRA supplementation led to a rapid increase in axial length, with thinner retina and sclera, without affecting refractive changes. The acceleration of axial elongation from RA supplementation differed from the ocular growth induced by FDM or LIM, as it was faster, greater in magnitude, and accompanied by choroidal thickening and flattening, corneal thickening, and lens thickness reduction [15,37]. Injecting the RAR antagonist LE540 into the vitreous body of guinea pigs induced by LIM partially inhibited the increase in RA levels and significantly slowed the development of myopia [43]. In guinea pigs, oral RA can attenuate the extent of short-wavelength inhibition of myopia, whereas oral administration of citral, an inhibitor of RA synthesis, slowed down eye growth [41]. Additionally, Summers Rada et al. identified RALDH2 as the key enzyme responsible for increased atRA synthesis in the choroid of chicks during periods of slowed eye growth (recovering from FDM) [44]. Expression of RALDH2 protein in the retina of guinea pigs induced by LIM also increased [14]. Three days after LIM induction, AHD2 (a synthetic enzyme of RA) mRNA levels in the retina were upregulated. And 6 h after positive lenses wearing, the abundance of retinal RA receptor RAR-β mRNA had already increased. RA synthesis inhibitor disulfiram can inhibit FDM [45].

Thus, we speculate on several mechanisms by which all-trans retinoic acid (atRA) might promote myopia progression:Extracellular matrix metabolism changes: Scleral thinning, reduced synthesis of scleral type I collagen, increased glycosaminoglycans, and matrix metalloproteinase 2 have been demonstrated to be associated with myopia development [46]. Experimental evidence suggests that exogenous atRA can upregulate metalloproteinase-2 in the sclera and downregulate collagen I and integrin β1 through the miR-328-PAX6 axis [47]. Moreover, RA has been shown to inhibit proteoglycan/GAG synthesis in the sclera of myopic monkeys and chickens [48]. Additionally, daily RA by gavage upregulates fibulin-1 in the sclera of guinea pigs, and in vitro, treatment with RA increases levels of Fbln1 in human scleral fibroblasts [49]. Fibulin-1 is an important extracellular matrix (ECM) molecule associated with matrix remodeling in many tissues [50]. The interaction between fibulin-1 and proteoglycans may be crucial, as the regulation of proteoglycan levels may play a key part in altering the scleral creep rate [51], thereby participating in axial elongation and influencing the development of myopia.Retinal level involvement: More studies have found that the rod pathway may be involved in the development of myopia. Clinically, it has been observed that peripheral defocused lenses could inhibit the development of myopia [52], while the peripheral retina was dominated by rods. Peripheral form deprivation in young monkeys could also induce FDM [53], and FDM could still be induced through the peripheral retina after laser ablation of the macula in young monkeys [54]. Knockdown of the rod pathway in mice resulted in refraction shifting toward hyperopia and failure to induce FDM [55], suggesting that the rod-dominated peripheral retina is involved in the development of myopia. Notably, vitamin A is also the material basis of the major photoreceptor pigment, rhodopsin, in rods [56]. Overexpression of rhodopsin in zebrafish promotes the development of myopia [57], whereas atropine inhibits myopia progression, likely by decreasing rhodopsin levels through increasing intraocular rhodopsin bleaching after pupil dilation. We, therefore, hypothesize that vitamin A may also be involved in myopia progression by affecting rhodopsin synthesis.Compensatory protective mechanism: Elevated levels of atRA in myopic eyes may serve as a compensatory protective mechanism: atRA may enhance the barrier function of the RPE in myopia by regulating the expression of intercellular tight junction-related proteins. Disrupted atRA signaling may lead to the destruction of functional tight junctions [42]. During the progression of myopia, the RPE layer undergoes stretching, and increased atRA levels can upregulate the expression of intercellular tight junction-related proteins, such as ZO-1 and occludin, which may help enhance the barrier function to counteract the tension of RPE cells [42]. However, with further development of myopia, axial lengthening of the eye can lead to loss of epithelial integrity, ultimately resulting in degeneration of the choroid and retina. Additionally, atRA promotes the secretion of TGF-β in RPE through the phospholipase C pathway [58].

These associations were estimated using independent data sources and combined in meta-analyses, ensuring sufficient statistical power and enhancing the robustness of the findings. Most observational studies included in regular meta-analyses are cross-sectional, which restricts their ability to establish causal relationships. However, the combination of meta-analysis and MR analysis with SNP as the IV helps to overcome this limitation. And MR research can be exempted from ethical review. The integration of MR analysis and meta-analysis in this study enhances causal inference of the association between vitamin A supplementation and risk of myopia, providing a comprehensive perspective. The MR approach is less subject to confounding or reverse causality bias than traditional studies [59].

However, this study has some limitations. One of the key limitations is the absence of specific data on the levels of dietary vitamin A intake and the amounts of vitamin A supplementation in the UK Biobank datasets. This limitation poses a significant challenge in interpreting our results, as the amount of vitamin A supplementation might critically influence the study outcomes. To address this issue, we refer to the data available in the European Food Safety Authority (EFSA) publication on Dietary Reference Values (DRVs) for vitamin A in Europeans. According to the EFSA, the recommended daily allowance (RDA) for vitamin A varies based on age, sex, and life stage, with adults recommended to consume 700–900 µg/day. However, actual intake levels can vary significantly among individuals and populations. In the absence of precise data from our datasets, it is crucial to consider these reference values and typical intake levels reported in other studies when interpreting our findings. The lack of detailed data on supplementation amounts means that we cannot ascertain whether participants were meeting, exceeding, or falling short of these recommended intake levels. Therefore, while our Mendelian randomization analysis suggests no causal association between genetic predisposition to vitamin A supplementation response and myopia, the absence of dosage data limits the robustness of this conclusion. Future research should aim to include more comprehensive dietary and supplementation data to provide a clearer picture of the relationship between vitamin A and myopia. This could involve detailed dietary assessments and accurate records of supplementation amounts, which would enable more precise evaluations of potential dose-response relationships.

In addition to this, considering that vitamin A supplementation is applicable to individuals of any sex, that the prevalence of myopia is not affected by sex [3], and that some of the databases on vitamin A supplementation we used were not stratified by sex, we did not stratify the MR analyses further by sex. To perform sensitivity analyses and horizontal pleiotropy testing, more genetic variants need to be included as IVs; therefore, the SNPs used in the analyses did not meet the traditional GWAS significance threshold (*p* < 5 × 10^−8^). In addition, the majority of participants in this study were of European origin, introducing potential confounding from population stratification. The specific range and distribution of both vitamin A supplementation and age of the included populations in the four databases that serve as the exposure were not given in their descriptions. But generally, the UK Biobank participants range from 40 to 69 years at recruitment, covering a wide adult age spectrum. Thus, it remains to be seen whether our findings can be extended to other racial groups. Future studies on the causal relationship between vitamin A supplementation and myopia should consider different European and non-European populations to achieve better generalizability.

## 5. Conclusions

In summary, the comprehensive results of this two-sample MR study and meta-analysis found no valid evidence supporting a direct association between vitamin A supplementation and myopia. Vitamin A supplementation may not have an independent effect on myopia, but notably, intraocular processes associated with vitamin A could potentially affect the development of myopia, suggesting an indirect association between vitamin A and myopia. Further studies are necessary to provide evidence of this association and elucidate its specific mechanisms, considering the low threshold settings for IV screening in our study compared to traditional thresholds and the ethnic heterogeneity of the study population.

## Figures and Tables

**Figure 1 nutrients-16-01933-f001:**
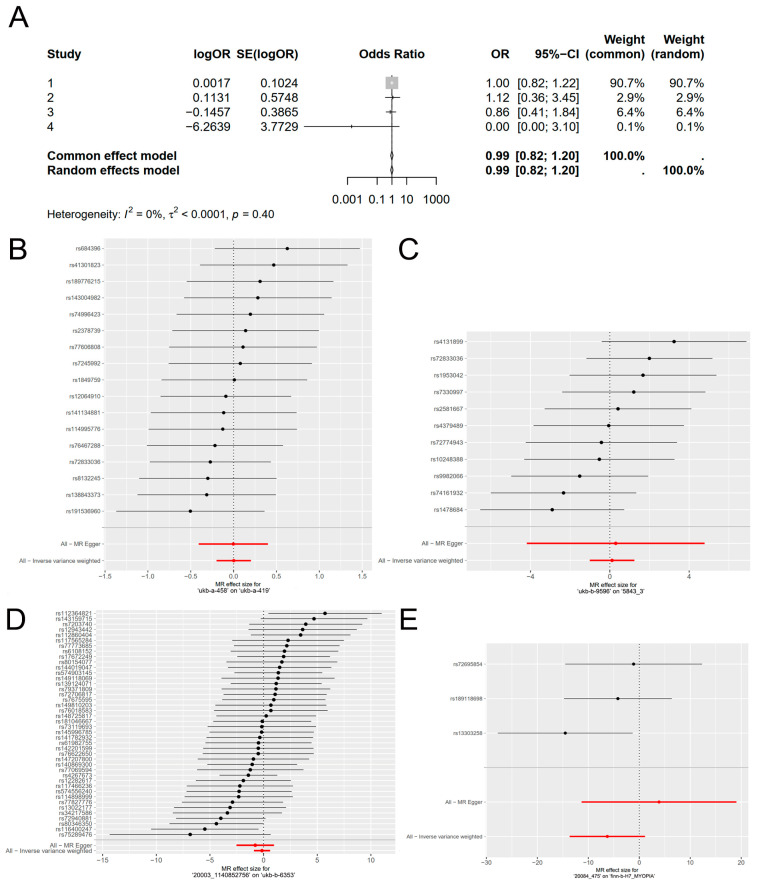
Associations of genetically predicted response to vitamin A supplementation with myopia. (**A**) Results of meta-analysis of (**B**–**E**). (**B**–**E**) The causal effect of exposure on outcome is individually estimated for each single nucleotide polymorphism (SNP) using the Wald ratio and presented in a forest plot. The MR estimate, incorporating all SNPs, is also provided using the MR-Egger and IVW methods. The data used were the exposure of ‘ukb-a-458′ on the outcome of ‘ukb-a-419′ (**B**), the exposure of ‘ukb-b-9596′ on the outcome of ‘5843_3′ (**C**), the exposure of ‘20003_1140852756′ on the outcome of ‘ukb-b-6353′, (**D**) and the exposure of ‘20084_475′ on the outcome of ‘finn-b-H7_MYOPIA’ (**E**).

**Figure 2 nutrients-16-01933-f002:**
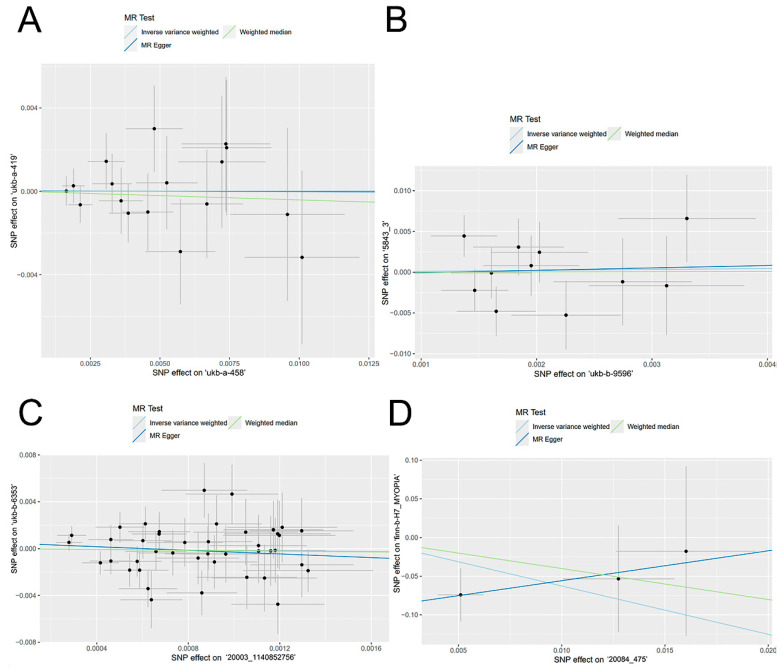
Scatterplots of the causal effect of vitamin A supplementation on myopia. The plotted data demonstrate the relationship between SNP effects on the outcome and SNP effects on the exposure, providing insights into the causal association. The data used were the exposure of ‘ukb-a-458′ on the outcome of ‘ukb-a-419′ (**A**), the exposure of ‘ukb-b-9596′ on the outcome of ‘5843_3′ (**B**), the exposure of ‘20003_1140852756′ on the outcome of ‘ukb-b-6353′ (**C**), and the exposure of ‘20084_475′ on the outcome of ‘finn-b-H7_MYOPIA’ (**D**).

**Figure 3 nutrients-16-01933-f003:**
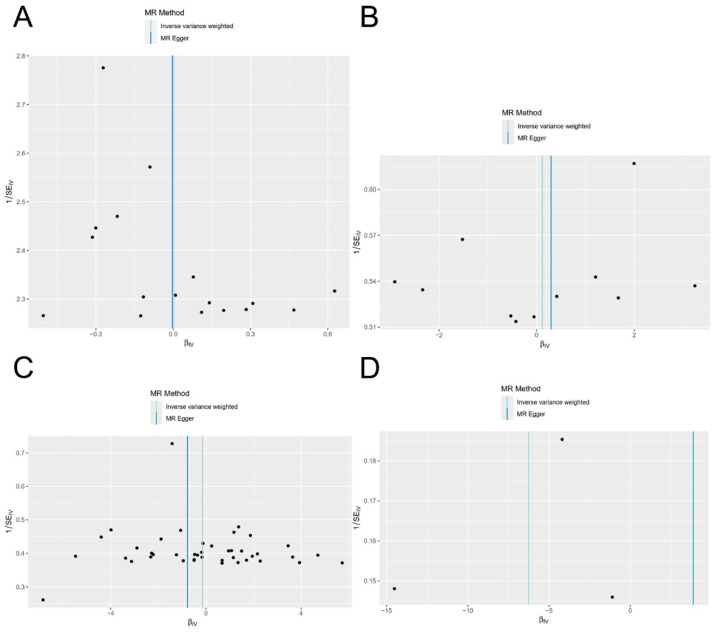
Funnel plot of Mendelian randomization. The data used were the exposure of ‘ukb-a-458′ on the outcome of ‘ukb-a-419′ (**A**), the exposure of ‘ukb-b-9596′ on the outcome of ‘5843_3′ (**B**), the exposure of ‘20003_1140852756′ on the outcome of ‘ukb-b-6353′ (**C**), and the exposure of ‘20084_475′ on the outcome of ‘finn-b-H7_MYOPIA’ (**D**).

**Figure 4 nutrients-16-01933-f004:**
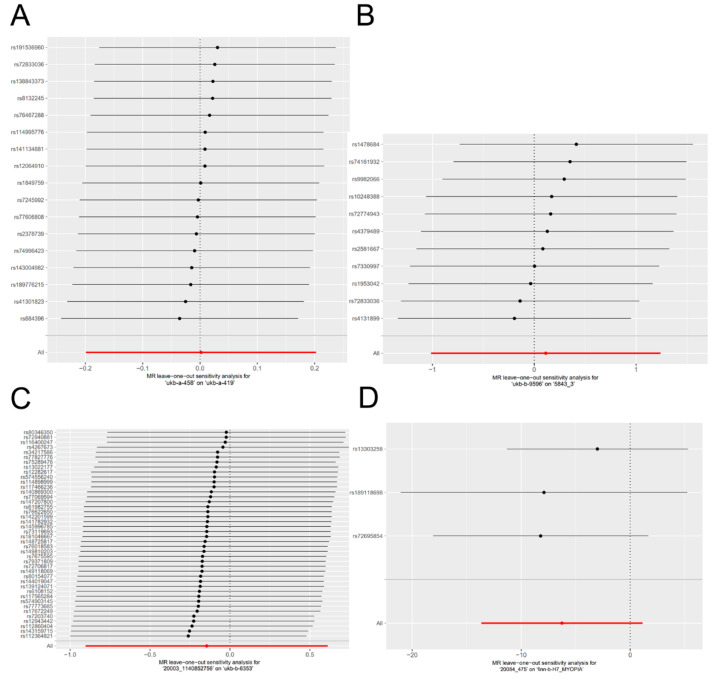
Leave-one-out plots for the causal association between vitamin A supplementation and myopia. Each point in the forest plot represents the MR analysis using IVW method, with the exclusion of the specific SNP associated with that data point. The data used were the exposure of ‘ukb-a-458′ on the outcome of ‘ukb-a-419′ (**A**), the exposure of ‘ukb-b-9596′ on the outcome of ‘5843_3′ (**B**), the exposure of ‘20003_1140852756′ on the outcome of ‘ukb-b-6353′ (**C**), and the exposure of ‘20084_475′ on the outcome of ‘finn-b-H7_MYOPIA’ (**D**).

**Table 1 nutrients-16-01933-t001:** Summary information on the GWAS database in the MR study.

Accession Number	Sample Size	Number of SNPs	Population Ethnicity	Gender	Comprehensive Database
ukb-a-458	335,591	10,894,596	European	Male and female	MRC IEU
ukb-b-9596	460,351	9,851,867	European	Male and female	MRC IEU
20003_1140852756	361,141	11,739,085	European	Male and female	UK Biobank
20084_475	51,427	11,731,938	European	Male and female	UK Biobank
ukb-a-419	335,700	10,894,596	European	Male and female	MRC IEU
5843_3	29,317	11,727,276	European	Male and female	UK Biobank
ukb-b-6353	460,536	9,851,867	European	Male and female	MRC IEU
finn-b-H7_MYOPIA	398,134	1,048,575	European	Male and female	FinnGen

**Table 2 nutrients-16-01933-t002:** MR estimates for the association between vitamin A supplementation and myopia.

Accession Number (Exposure, Outcome)	SNP	Method	β	OR (95%CI)	*p*
ukb-a-458, ukb-a-419	17	MR-Egger	−0.003	0.997 (0.665, 1.493)	0.987
		Weighted median	−0.041	0.960 (0.740, 1.245)	0.757
		Inverse variance weighted	0.002	1.002 (0.820, 1.224)	0.987
		Simple mode	−0.078	0.925 (0.580, 1.475)	0.748
		Weighted mode	−0.134	0.874 (0.546, 1.401)	0.585
ukb-b-9596, 5843_3	11	MR-Egger	0.295	1.344 (0.015, 119.070)	0.900
		Weighted median	0.017	1.018 (0.211, 4.910)	0.983
		Inverse variance weighted	0.113	1.120 (0.363, 3.455)	0.844
		Simple mode	0.106	1.112 (0.079, 15.747)	0.939
		Weighted mode	0.228	1.256 (0.082, 19.281)	0.873
20003_1140852756, ukb-b-6353	42	MR-Egger	−0.778	0.460 (0.080, 2.646)	0.389
		Weighted median	−0.171	0.843 (0.287, 2.475)	0.756
		Inverse variance weighted	−0.146	0.865 (0.405, 1.844)	0.706
		Simple mode	0.573	1.774 (0.150, 21.016)	0.652
		Weighted mode	0.313	1.368 (0.153, 12.247)	0.781
20084_475, finn-b-H7_MYOPIA	3	MR-Egger	3.875	48.163 (1.21 × 10^−5^, 1.92 × 108)	0.705
		Weighted median	−4.007	0.018 (1.84 × 10^−6^, 179.916)	0.393
		Inverse variance weighted	−6.264	0.002 (1.17 × 10^−6^, 3.099)	0.097
		Simple mode	−2.639	0.071 (1.97 × 10^−6^, 2591.787)	0.671
		Weighted mode	−3.095	0.045 (7.76 × 10^−7^, 2640.854)	0.636

Two outliers were identified in one of these groups while performing the MR-PRESSO analysis. After removing these two outliers, the MR-PRESSO analysis did not find any significant outliers in the four sets of outcomes to ensure no horizontal pleiotropy.

**Table 3 nutrients-16-01933-t003:** MR heterogeneity analysis results.

Accession Number (Exposure, Outcome)	Method	Q	*p*	*I* ^2^
ukb-a-458, ukb-a-419	MR-Egger	8.048	0.922	0.864
	Inverse variance weighted	8.049	0.947	0.988
ukb-b-9596, 5843_3	MR-Egger	10.637	0.301	0.154
	Inverse variance weighted	10.645	0.386	0.061
20003_1140852756, ukb-b-6353	MR-Egger	43.249	0.334	0.075
	Inverse variance weighted	43.916	0.349	0.066
20084_475, finn-b-H7_MYOPIA	MR-Egger	0.033	0.856	29.378
	Inverse variance weighted	2.209	0.331	0.095

**Table 4 nutrients-16-01933-t004:** MR pleiotropy analysis results.

Accession Number (Exposure, Outcome)	Intercept	SE	*p*
ukb-a-458, ukb-a-419	2 × 10^−5^	0.001	0.977
ukb-b-9596, 5843_3	3.6 × 10^−4^	0.004	0.936
20003_1140852756, ukb-b-6353	4.8 × 10^−4^	0.001	0.437
20084_475, finn-b-H7_MYOPIA	−0.095	0.064	0.379

## Data Availability

The datasets analyzed during the current study are available in the IEU Open GWAS project (https://gwas.mrcieu.ac.uk/, accessed on 27 February 2024), the UK Biobank (http://www.nealelab.is/uk-biobank, accessed on 27 February 2024), FinnGen (https://www.finngen.fi/fi, accessed on 27 February 2024).

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
