# Peer review of "No Evidence of an Association between Genetic Factors Affecting Response to Vitamin A Supplementation and Myopia: A Mendelian Randomization Study and Meta-Analysis"

_nutrients, 2024, doi:10.3390/nu16121933_

Round 1

Reviewer 1 Report

Comments and Suggestions for Authors

The manuscript entitled “No Evidence of a Genetic Causal Association between Vitamin A Intake and Myopia A Mendelian Randomization Study and Meta-analysis” presents interesting issue but some problems must be corrected before publishing.

Major:

Authors use the term ‘vitamin A intake’ but also ‘vitamin A supplement’, and they do not explain what in fact did they analyse – either dietary intake (from a diet), or supplementation (from dietary supplements). Or maybe both – from a diet and supplements combined. Authors must clearly explain in and they must stay consistent – the included data must be based on the same variable – either dietary intake (from a diet), or supplementation (from dietary supplements).

General:

Authors should prepare their manuscript according to the instructions for Authors – it seems that the study was rejected from the other journal and submitted to Nutrients without adjusting it to recommendations of the journal.

Abstract:

The abstract should be a single paragraph and should follow the style of structured abstracts, but without headings

Introduction:

Authors should deepen the presentation of the associations between vitamin A (and other vitamins – as they have mentioned that for the other vitamins there are also such associations but they did not describe them) and myopia. In the current version of the manuscript Authors mentioned only one study presenting a weak association. It is not a strong justification for the presented study.

If Authors did not observe the association in their study and they did not present adequate justification of the study, we may suppose that there was no sense to conduct the analysis as there was no reason to suppose that there is any association. Authors should present adequate literature background.

Materials and Methods:

It should be clearly described how was vitamin A intake associated within the used databases. Especially for vitamin A (as there are numerous forms of this vitamin), Authors should clearly describe the applied method and if it was coherent.

Results:

The description of the results is incoherent – as described above, we even do not know what variable is studied - dietary intake (from a diet), or supplementation (from dietary supplements).

There is no sense to present multiple analyses if they are not properly planned and based on a justified scientific aim.

Discussion:

The discussion should be based on the conducted study (including what was studied either dietary intake (from a diet), or supplementation (from dietary supplements)).

Conclusions:

The limitations of the study should be highlighted within this section.

Reviewer 2 Report

Comments and Suggestions for Authors

The manuscript “No Evidence of a Genetic Causal Association between Vitamin A Intake and Myopia A Mendelian Randomization Study and Meta-analysis” by Xu is a research article which examined the influence of vitamin A on the risk of myopia using Mendelian randomization (MR) and meta-analytical techniques. In this study, genetic variants from the UK Biobank and FinnGen studies associated with vitamin A intake were used as instrumental variables to assess the causal relationship between vitamin A and myopia. The authors found that the meta-analysis of MR results did not reveal any valid evidence supporting a direct causal association between vitamin A intake and myopia risk. However, the other set of results suggested that vitamin A intake was a protective factor for myopia. Therefore, the authors concluded that Vitamin A consumption may not be an independent influence on myopia, but intraocular processes associated with vitamin A may indirectly influence the onset and development of myopia. In general, this review article is critical in this field and contains essential contents. However, I have several comments before this manuscript is accepted for publication.

1. Have the authors examined the difference between men and female?

2. Is there any reference which shows that intraocular processes associated with vitamin A may influence the development of myopia? Please add the references!
